

# Survival of side grafts with scions from pure species *Pinus engelmannii* Carr. and the *P. engelmannii* × *P. arizonica* Engelm. var. *arizonica* hybrid

Alberto Pérez-Luna[1], Christian Wehenkel[2], José Ángel Prieto-Ruíz[3], Javier López-Upton[4] and José Ciro Hernández-Díaz[2]

[1] Programa Institucional de Doctorado en Ciencias Agropecuarias y Forestales, Universidad Juárez del Estado de Durango, Durango, México
[2] Instituto de Silvicultura e Industria de la Madera, Universidad Juárez del Estado de Durango, Durango, México
[3] Facultad de Ciencias Forestales, Universidad Juárez del Estado de Durango, Durango, México
[4] Campus Montecillo, Colegio de Posgraduados, Texcoco, Estado de México, México

Corresponding author
José Ciro Hernández-Díaz,
jciroh@ujed.mx

## ABSTRACT

Grafting is one of the most widely used methods for vegetative propagation, particularly for multiplying trees considered important, but there has been little research done on the effect of hybridization on grafts from the genus *Pinus*. Sometimes hybrids show the ability to reproduce and adapt efficiently to the environment. However, they reduce the genetic gain of seed orchards. The objective in this research was to evaluate the effect of scion grafts from pure species donor *Pinus engelmannii* Carr. and from putative hybrid trees *P. engelmannii* × *P. arizonica* Engelm., grafted on rootstocks of pure species *P. engelmannii*, along with the effect of the position of the scion in the donor tree crown (upper third and middle third). The scions were collected from three trees of the pure species and three hybrids. In each tree, 20 scions were collected from each third of the crown evaluated. 120 side-veneer grafts were made at the beginning of spring (March) 2018. Variance analyses were performed to evaluate the treatments and adjustments of the Logit and Weibull models to obtain the probability of graft survival. Significant differences were found between the origins of scions ($p < 0.0083$, after Bonferroni correction), showing grafts with hybrid tree scions taking hold better. In addition, the probability of survival at 5 months after grafting with hybrid tree scions was greater ($p < 0.0001$) than in grafts with scions from trees of the pure species (Logit model), which coincides with the results of the Weibull model, which indicated that the probability of graft death with pure species donor tree scions is greater than for grafts with hybrid scions. There were no significant differences regarding the position of the scion in the donor tree crown.

## INTRODUCTION

Between 1990 and 2010, México was one of seven countries with the highest annual net loss of forest area, with an approximate deforestation rate of 500,000 ha per year

(*FAO-CONAFOR, 2009*; *Rosete-Vergés et al., 2014*). In Durango, México, from 1986 to 2012, a total area between 886,679 and 1,398,328 hectares of temperate forest cover were lost due to different types of disturbances, which represented 29–34% of the total forest area in the State (*Novo-Fernández et al., 2018*). This growing degradation of Mexican forests, caused by the loss of temperate forest cover, contributes to the annual deficit of approximately 30 million cubic meters of wood, which is due to the increase in national consumption of this raw material and the 20% increase in demand for forest products in North America (*Fiedler et al., 2001*; *Chidumayo & Gumbo, 2013*; *SEMARNAT (Secretaría de Medio Ambiente y Recursos Naturales), 2016*). It is necessary to promote genetic improvement programs that allow the safeguarding, propagation and improvement of genotypes of high ecological, commercial and cultural value (*Neale & Kremer, 2011*; *Wheeler et al., 2015*; *Gray et al., 2016*). Likewise, genetic improvements may allow for the obtaining of better survival and productivity results in commercial forest plantations, which in México have an approximate annual mortality rate of 45% (*Muñoz Flores et al., 2012*; *Rosales Mata et al., 2015*).

In northern México and the southern United States of America, *Pinus engelmannii* and *P. arizonica* Engelm var. *arizonica* are two of the most important conifers from an ecological and commercial standpoint (*Barton, Swetnam & Baisan, 2001*; *García Árevalo & González Elizondo, 2003*; *Ávila-Flores et al., 2016a*; *Friedrich et al., 2018*). In addition, *P. engelmannii* is one of the most utilized species in northern México in commercial forest plantations, since this species' wood has a high demand and economic value (*Prieto Ruíz et al., 2004*). *P. engelmannii* trees can reach up to 35 m in height and up to 100 cm in diameter (*Ávila-Flores et al., 2016a*, *2016b*). *P. arizonica* trees can reach up to 25 m in height and 80 cm in diameter, with a straight trunk and a natural self-pruning that helps reduce mechanical and asthetic defects in the wood (*Barton, Swetnam & Baisan, 2001*; *García Árevalo & González Elizondo, 2003*). *P. engelmannii* needles present 6–23 stomata rows on the dorsal side and 5–9 rows on the ventral side; while *P. arizonica* needles have 3–8 rows on the dorsal side and 3–7 rows on the ventral side (*García Árevalo & González Elizondo, 2003*). *Yeaton, Yeaton & Waggoner (1983)* described that pine species trees from Durango, México, that have fewer stomata rows on the dorsal and ventral sides of the needles, are more stress-tolerant and have a tendency to grow in semi-arid environments.

The genetic improvement of forest species is mainly carried out in asexual seed orchards (ASO) and in clone banks (CB). In ASO, genetically enhanced seed is produced through open or controlled pollination (*Stewart et al., 2016*) and in CB, it is possible to produce identical clones through vegetative propagation (*Mutke, Gordo & Gil, 2005*; *Şevik & Topaçoğlu, 2015*; *Oliveira, Nogueira & Higa, 2018*). Grafting is the most widely used method of vegetative propagation to establish ASO and CB in the genus *Pinus* (*Jayawickrama, Jett & McKeand, 1991*; *Stewart et al., 2016*; *Vargas Hernández & Vargas Abonce, 2016*; *Pérez-Luna et al., 2019*).

A graft is the union of a rootstock and a scion of different origin, which form a new combination of cells that gives rise to a new plant (*Mudge et al., 2009*; *Pérez-Luna et al., in press*). Grafts can be intraspecific (scion and rootstock of the same species) or

interspecific (scion and rootstock of different species) (*Opoku, Opuni-Frimpong & Dompreh, 2018*), and both types have been used in coniferous grafting (*Lott et al., 2003*; *Barnes, 2008*). The most commonly used grafting techniques in *Pinus* species are top cleft and side-veneer (*Staubach & Fins, 1988*; *Cuevas Cruz, 2014*; *Pérez-Luna et al., 2019*). Although in many cases coniferous grafting has been successfully achieved (*Hibbert-Frey et al., 2011*; *Almqvist, 2013a*), in *P. engelmannii* Carr., the survival rate has been low (*Pérez-Luna et al., 2019*).

The following are the main factors involved in the success rate of coniferous and other vegetal species grafting: The technique used in grafting, the skill level of the person doing the grafting, rootstock quality, scion's vigor, origin of scions and rootstocks (*Jayawickrama, Jett & McKeand, 1991*; *Lott et al., 2003*; *Pérez-Luna et al., in press*), position of the scions in the crown of donor trees, the age of scion donor trees, genetic, taxonomic and anatomical characteristics (*Lott et al., 2003*; *Hibbert-Frey et al., 2011*; *Pérez-Luna et al., 2019*), cell compatibility of the scion with the rootstock, the correct connection of cell tissue and the presence or absence of growth regulators hormones (gibberellins, auxins and cytokinins, for example) (*Valdés, Centeno & Fernández, 2003*; *Pina & Errea, 2005*; *Moore, 1984*).

Another factor that can influence the success of grafting is the amount of resin channels in parent species. *P. engelmannii* has more resin channels than *P. arizonica* (*García Árevalo & González Elizondo, 2003*), and *Pérez-Luna et al. (2019)* found that a larger amount of resin channels in the scion lead to a smaller percentage of graft compatibility.

Elongation of grafts can be observed in the first or second month after grafting (*Pérez-Luna et al., 2019*). Flowering can occur on the grafts 2 years after grafting has taken place. However, the seed production time of grafts in *Pinus* species can vary from 5 to 7 years, depending on the species (*Luukkanen & Johansson, 1980*; *Pijut, 2002*). One method that is used to accelerate seed production in grafts is the application of hormones, such as gibberellins (*Hare, 1984*; *Pijut, 2002*).

Although grafts may sometimes show adequate initial growth and flowering (*Pérez-Luna et al., 2019*), mortality can occur due to possible incompatibility between the scion and the rootstock after germplasm production, even 5 years or more after grafting (*Jayawickrama, Jett & McKeand, 1991*; *Valera et al., 1999*; *Valdés, Centeno & Fernández, 2003*). The origin of this incompatibility can be genetic and/or anatomical (*Lott et al., 2003*; *Pérez-Luna et al., 2019*).

Using hybrids can be an alternative to improve commercial plantations and restore degraded forest ecosystems (*Lopez et al., 2018*; *Skousen et al., 2018*). Natural hybridization between *Pinus* species can be detected through morphological analysis (*García Árevalo & González Elizondo, 2003*), but it is better to apply molecular markers techniques, such as AFLP (*Ávila-Flores et al., 2016a*; C. Wehenkel et al., 2020, unpublished data). In certain regions of the state of Durango, putative hybrids have been observed as a product of natural cross between *Pinus* species, such as *P. engelmannii × P. arizonica*, which have adapted to sites with adverse climatic conditions (*Ávila-Flores et al., 2016a*).

In grafting, the use of hybrids has been studied in conifer species (*Marchal et al., 2017*; *Kita et al., 2018*), however, in pines, it has not been studied much (*Lott et al., 2003*). In contrast, some plants, such as tomato and watermelon have been studied extensively, and survival levels greater than 80% have been reported (*Djidonou et al., 2016*; *Xu et al., 2016*; *Zhang et al., 2019*).

Survival models are very useful to make predictions about the probability of survival and death through the time, as a function of two or more treatments (*Kleinbaum & Klein, 2002*; *Zhang, 2016*). The development of models, such as the Cox proportional hazards model, Weibull accelerated failure time model, Weibull risk function, as well as the Logit model, have had great application in clinical analyzes (*Sullivan, Massaro & D'Agostino, 2004*; *Vazquez et al., 2009*; *Chaou et al., 2017*). However, they have been scarcely used in forestry research (*Pérez-Luna et al., 2019*).

The objective of this study was to determine the survival in *P. engelmannii* grafting by using scions, both from pure species donor trees and putative hybrids identified with AFLP, as well as the influence of the position of the scion in the donor tree crown (upper third and middle third). The rootstocks were produced using seeds from a stand that was identified as pure species *P. engelmannii*, on the basis of its morphological characteristics. The possible effect of six graft variables of the scions and rootstocks on the probability of survival, at 5 months after grafting, was also evaluated. In this study, we successfully applied several of the models mentioned above, which can also be useful in enriching statistical analysis in future forestry research.

# MATERIALS AND METHODS

## Place of origin and preparation method of rootstocks

The seed to produce the rootstocks was collected in a natural stand of *P. engelmannii* trees located in Empalme Purísima, municipality of Durango, Dgo., México, at coordinates 23°55′48.1″N and 105°05′43.0″W, at 2,480 m elevation. In this stand, no morphological evidence of natural hybridization was observed, for which reason the rootstocks produced were considered pure trees of said specie. For 11 months the rootstocks grew in 77-cavity, expanded polystyrene seedling trays with 170 mL per cavity. Afterwards, the plant grew for 19 months in black polyethylene 5 L bags (20 cm in width and 30 cm in length). At the time of grafting, the rootstocks were 2.5 years old. During this period, the plants reached an average neck diameter of 18.4 mm and an average height of 17.4 cm.

## Selection of donor trees and collection of scions

The scions were collected in a free access natural stand located in the area known as El Río, in the Ejido La Casita, municipality of Durango, Dgo., México, at coordinates 23°43′26.8″N and 104°47′27.1″W, at an elevation of 2,335 m. A permit for the collection of samples in the field was not required, since the species studied are not under any conservation status. The scions were collected on March 15, 2018, from a seed producer stand consisting of both tree types, pure species *P. engelmannii* and *P. engelmannii* × *P. arizonica* hybrids, detected by AFLP and morphological traits (analyzed by *Ávila-Flores et al., 2016a*).

Scions' donor trees were divided in the two categories of hybrids and trees of the pure species, based on the membership probabilities (MP) of each individual to pure *P. engelmannii* (pure trees corresponded to an MP larger than 0.945, while the hybrids' MP was smaller than 0.610). Three trees were chosen from each category; the growth and vigor of each was similar between these two categories within the stand (*Ávila-Flores et al., 2016a*).

The average age of the three trees of the pure species was 37 years, with a height of 13 m and diameter of 30 cm on average. The average age of the hybrid trees also was 37 years, with a height of 14.5 m and a diameter of 36 cm on average. During the collection of the donor trees scions, selected from the pure species of *P. engelmannii* and *P. engelmannii* × *P. arizonica* hybrids, the following morphological differences were detected between the studied groups of trees: Conical crown in the trees of the pure species and irregular shaped crown in the hybrids trees, slightly hanging branches in the lower part of the hybrid tree crowns, while more erect branches in the whole crown of the trees of the pure species (Figs. 1A and 1B). The cone size of the *P. engelmannii* × *P. arizonica* hybrids was smaller than the cones of the pure species *P. engelmannii* (Figs. 1C and 1D). Erect fascicles in the trees of the pure species and fallen fascicles in hybrids (Figs. 1E and 1F).

From the crown of each donor tree, 20 scions were collected from the upper third and 20 from the middle third. Each scion had an approximate length of 20 cm. In order to safely transport the scions to the grafting site, they were packed in rectangular 72 L plastic boxes containing sawdust moistened with a solution of Captán® fungicide at a dose of 3 gL$^{-1}$, to prevent the drying of the scions and their contamination by fungi. Of the 20 scions collected from each tree, only 10 were selected to be grafted. The selection of these scions was made according to their possibility of diameter size tie with the rootstocks.

## Grafting

The grafting was done in the nursery at the Instituto de Silvicultura e Industria de la Madera de la Universidad Juárez del Estado de Durango (ISIMA-UJED). The grafts were placed in a greenhouse of 6 × 8 × 3 m of width, length and height, respectively, with a white plastic cover. To reduce the temperature inside the greenhouse, two shade meshes were placed above the roof, with an average spacing of 30 cm between them. The upper mesh had 70% light retention and the lower mesh had 50% light retention.

A total of 120 grafts were made using *P. engelmannii* rootstocks; 60 of these grafts were done with scions from *P. engelmannii* × *P. arizonica* hybrid donor trees, and the other 60 with scions from pure species of *P. engelmannii* trees. The grafting was done on March 16, 2018, 1 day after the scions were picked. The side-veneer grafting technique described by *Muñoz Flores et al. (2013)* and *Pérez-Luna et al. (2019)* was used.

The graft substrate was watered every 3 days and as a complement in each irrigation, triple 19 water-soluble fertilizer (N-P-K) was applied at a dose of 3 gL$^{-1}$. Fertigation was carried out during the 5 months of evaluation, using a 7 L manual watering can. In addition, from the 2nd month after grafting (May 2019) Promyl® fungicide was applied to plants' substrates through irrigation at a dose of 2 gL$^{-1}$ to prevent fungal damage. This activity was carried out every 8 days for a month.

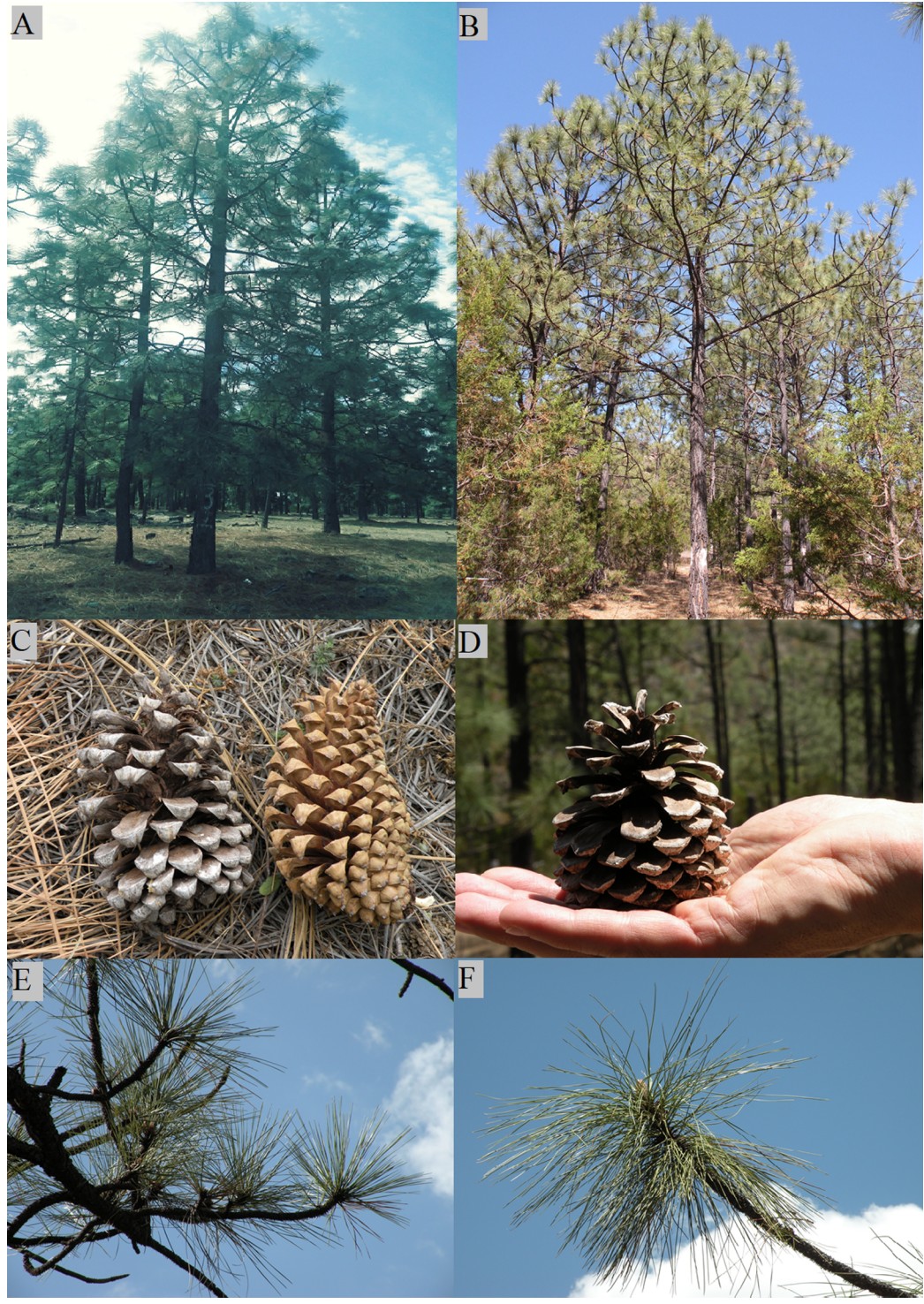

**Figure 1 Morphological characteristics of pure species of *Pinus engelmannii* and *P. engelmannii* × *P. arizonica* hybrid.** (A) Tree of the pure species; (B) hybrid tree; (C) cones of pure species; (D) cone of hybrid; (E) fascicles of pure species; (F) fascicles of hybrid. Photo credit: M. Socorro González Elizondo.

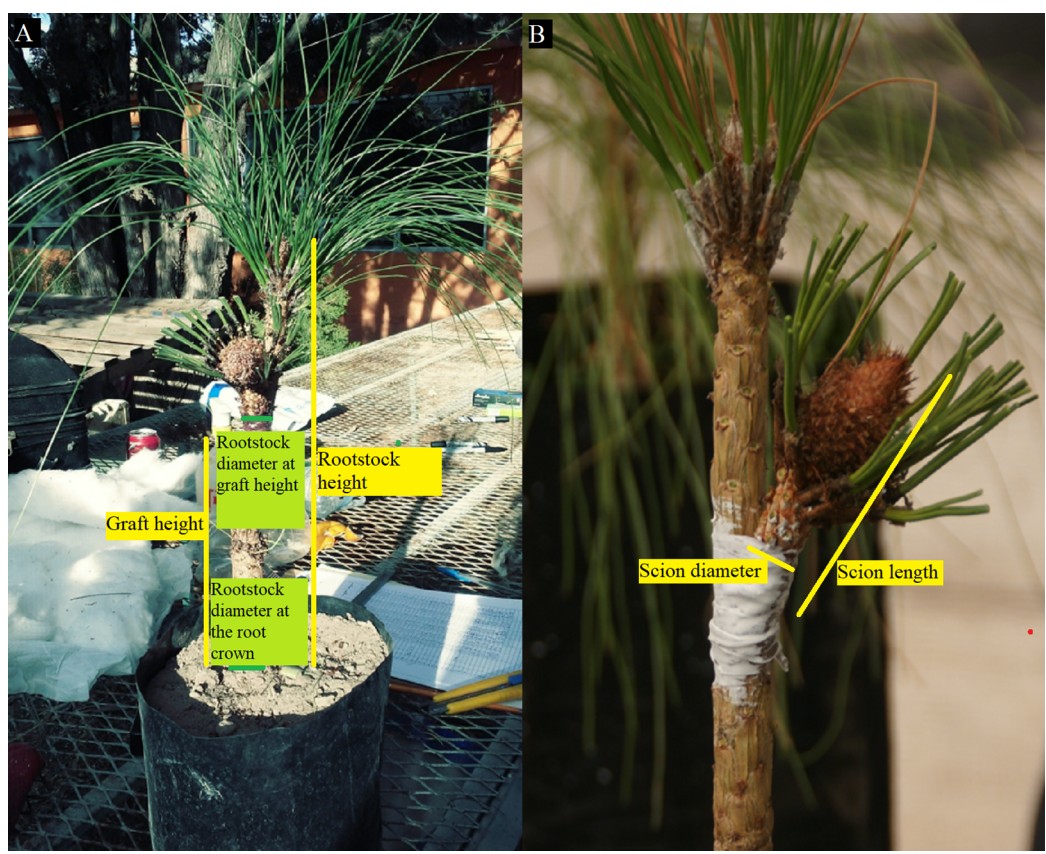

**Figure 2** **Measured sections to obtain the graft variables in side veneer grafts of *Pinus*.** (A) Variables in rootstocks and grafts; (B) variables in scions. Photo credit: Alberto Pérez Luna.

Full-size ⟨image⟩ DOI: 10.7717/peerj.8468/fig-2

## Treatments, variables evaluated and statistical analysis

Four treatments were evaluated: Two types of donor trees (*P. engelmannii* × *P. arizonica* hybrids and pure species *P. engelmannii*) and two positions of the scions in the donor tree crown (middle and upper third). Each graft represented an experimental unit, and there were 30 repetitions for each treatment, distributed under a randomized block arrangement.

Before grafting, the following graft variables corresponding to the scions and rootstocks were recorded: Scion length (SL), scion diameter (SD), rootstock height (RH), rootstock diameter at the root crown (RD), graft height (GH) and RD at graft height (GD). The sections of scions and rootstocks measured for obtaining the grafts variables are described in Fig. 2.

An analysis of variance (one-way ANOVA) was performed in the R program (*R Core Team, 2013*), finding significant differences between the pure species and hybrid donor trees in regard to the values of each analyzed graft variable (Table 1).

After grafting, graft survival was evaluated as a response variable each month for 5 months. As independent variables, the type of scion donor tree (hybrid or tree of the pure species) (TDT), the position of the scion in the donor tree crown (SPC) and the graft
**Table 1 Values of the analyzed variables in grafts with scions from trees of the pure species of *Pinus engelmannii* and from *P. engelmannii* × *P. arizonica* hybrids, on *P. engelmannii* rootstocks.**

| Graft variable | Minimum | Mean ± standard deviation | Maximum | Variation coefficient | p-value one-way ANOVA |
|---|---|---|---|---|---|
| Scion length (SL-cm) | 4.0 | 8.1 ± 2.0 | 15.0 | 0.25 | <0.0001 |
| Scion diameter (SD-mm) | 8.0 | 13.5 ± 2.4 | 21.5 | 0.18 | <0.0001 |
| Rootstock height (RH-cm) | 11.0 | 17.4 ± 3.9 | 30.0 | 0.22 | <0.0001 |
| Rootstock diameter at the root crown (RD-mm) | 11.0 | 18.4 ± 3.6 | 27.5 | 0.19 | <0.0001 |
| Graft height (GH-cm) | 3.0 | 7.7 ± 2.4 | 15.0 | 0.30 | <0.0001 |
| Rootstock diameter at graft height (GD-mm) | 9.6 | 15.4 ± 2.9 | 24.1 | 0.18 | <0.0001 |

variables obtained from the scions and rootstocks were considered. The normality of the graft variables was evaluated with the Kolmogorov–Smirnov test. To evaluate the effect of the treatments on survival, an analysis of variance and a comparison test of means, with a proposed initial value of $\alpha = 0.05$, were performed with the data obtained in each month.

Before carrying out this test and in order to reduce the probability of committing a type I error, the Bonferroni correction was applied to adjust the level of significance (*Weisstein, 2004*), resulting in the corrected $\alpha$ value = 0.0083, which was obtained by dividing the initial value of alpha by the number of parallel comparisons (0.05/6 = 0.0083). The same adjustment and corrected alpha value (0.0083) was used in the analysis of all the survival models described below.

Also, a Pearson's correlation test was performed to detect the possible multicollinearity between the analyzed graft variables, and thus be able to discriminate the redundant variables in the adjustment of the evaluated survival models. In addition, two models were adjusted to determine the probability of graft survival based on the treatments evaluated. These models were the Logit function and the Weibull accelerated failure time model. All the statistical tests described were performed on the free software platform R (*R Core Team, 2013*).

## Adjustment in the logit model

To assess the effect of treatments and graft variables on the probability of occurrence of successful events (graft survival) at the end of the evaluation period, the non-parametric Logit model (*Hastie & Tibshirani, 1987*; *Kleinbaum & Klein, 2002*), was adjusted with the "aod" package of the R platform (*R Core Team, 2013*). The Logit model is defined by the following equation:

$$p(x_i) = \frac{1}{1 + e^{-(\alpha + \sum \beta_i x_i)}} \tag{1}$$

Where: $p(x_i)$ is the probability of occurrence of an event to be evaluated (in this case the survival of grafts) of the response variable, which is inversely proportional to an exponential function elevated to the sum of the intercept $\alpha$ and $n$ predictors $x$. The Logit model works with dichotomous (dummy) and discrete variables. Likewise, the adjustment of the Logit model generates estimators of the $\beta$ coefficients for the predictors; if the

value of the estimator is positive, the probability of occurrence of the successful event ($p(x_i)$) increases when the value of the respective independent variable ($x_i$) increases by one unit, and if the estimator is negative, the probability of the successful event decreases by increasing the value of the independent variable by one unit (*Kleinbaum & Klein, 2002*).

Furthermore, the Logit model generates the odds ratio that allows determining the probability of occurrence of the event to be evaluated (survival rate) (*Bland & Altman, 2000*; *Kleinbaum & Klein, 2002*; *Aedo, Pavlov & Clavero, 2010*). The equation to calculate the odds ratio is:

$$\text{Odds ratio} = \frac{p(x_1)(1 - p(x_0))}{p(x_0)(1 - p(x_1))} \qquad (2)$$

Where $p(x_1)$ is the probability of the occurrence of the graft survival event with hybrid donor tree scions and $p(x_0)$ is the probability of graft survival with pure species donor tree scions. The interpretation of the odds ratio is always relative, since its value is dimensionless and is obtained by dividing the product of multiplying the probability of graft survival with hybrid tree scions ($p(x_1)$) by the probability of graft mortality with scions from trees of the pure species ($1 - p(x_0)$), divided by the product of multiplying the probability of graft survival with scions from trees of the pure species ($p(x_0)$) by the probability of graft mortality with hybrid tree scions ($1 - p(x_1)$) (*Bland & Altman, 2000*; *Kleinbaum & Klein, 2002*; *Aedo, Pavlov & Clavero, 2010*).

To calculate the odds ratio, the most successful treatment of the event to be evaluated is used as a reference; that is, if the odds ratio of survival is desired to be obtained, the first factor of the numerator (Eq. 2), will be the $p(x_i)$ of the treatment with the highest proportion of living specimens, which in this study was the graft of hybrid tree scions ($x_1 = 1$), that is to say $p(x_1)$ and therefore the first denominator factor will be $p(x_0)$, which is the proportion of live grafts coming from the other treatment (scions from trees of the pure species).

However, the odds ratio can also be calculated in another way, through an operation that involves the relative frequencies of the living and dead specimens of each treatment under study (Eq. 3) (*Kleinbaum & Klein, 2002*).

$$\text{Odds ratio} = \frac{a \times d}{b \times c} \qquad (3)$$

Where: $a$ is the relative frequency of hybrid tree scion grafts ($x_1 = 1$) that are alive at the end of the evaluation period and $c$ is the relative frequency of grafts with scions from trees of the pure species ($x_0 = 0$) that are found alive at the end of that period, while $b$ and $d$ are the relative frequencies of dead grafts of hybrid tree scions ($x_1 = 1$) and scions from trees of the pure species ($x_0 = 0$), respectively.

In the study, graft survival was considered as the event to be evaluated. To evaluate it, dead grafts were coded with zero (0), while live grafts were coded as one (1), in order to obtain the odds ratio of the most successful treatment (survival) and contrast it with the treatment of least survival. In the case of the variables referring to the type of scion donor tree and the scion position in the donor tree crown, the following dichotomous
coding was used: grafts with pure species tree scions (0) and grafts with hybrid tree scions (1); grafts with scions from the upper third of the crown (0) and grafts with scions from the middle third of the crown (1).

## Adjustment of the Weibull accelerated failure time model and the Weibull risk function

Graft survival was analyzed with a Weibull accelerated failure time model (AFT) under a two-parameter Weibull distribution to obtain the graft mortality risk ratio (hazard ratio) based on the evaluated treatments and variables. The AFT model was adjusted with the R "survival" library (*R Core Team, 2013*). For this, the variables TDT (type of donor tree), SPC (scion position in the crown) and the graft variables were used as predictor variables. The AFT model is a logarithmic-linear variant of the Weibull distribution model (*Van Houwelingen, 2000*), and allows estimating the effect of a variable or set of variables on the duration of survival, which helps to estimate the average time (T) in which there is at least one failure (illness or death) (*Carroll, 2003*; *Zhang, 2016*). This model is defined as:

$$In(T) = \alpha + \delta x + \sigma\varepsilon \tag{4}$$

Where: $In(T)$ is the natural logarithm of the average time ($T$) in which it is estimated that there will be at least one mortality (failure) event among the evaluated variables (live grafts), $\alpha$ is the scale parameter, $\delta$ it is the coefficient of the explanatory variable, which in the AFT model is known as the "acceleration factor," $\sigma$ is the shape parameter of the model and $\varepsilon$ is the error of the distribution function (*George, Seals & Aban, 2014*). All these parameters were estimated with the "*survival*" library of the R program (*R Core Team, 2013*). In the AFT model, if $\delta < 0$ the survival time decreases due to the effect of the variable $x$, if $\delta = 0$ the effect of $x$ is constant, so the survival time does not increase or decrease due to the effect of any change in the value of the variable $x$, and if $\delta > 0$ the survival time increases due to the effect of the variable $x$. To interpret the value of $\delta$ linearly, it is necessary to apply this coefficient as an exponent, that is, $e^{(\delta)}$ (*Qi, 2009*). It is important to mention that if $\delta < 0$ the value of $\alpha$ increases, and if $\delta > 0$ the value of $\alpha$ decreases, keeping the value of $\sigma$ constant in both cases. Therefore, the effect of the predictor variable $x$ is multiplicative (logarithmic) on the time scale, so it is said that this model accelerates (increases or decreases) the estimated average survival time of the specimens studied up to time $T$, which depends on the effectiveness or ineffectiveness of the treatments evaluated. Said time $T$ is always longer than the duration of the evaluation (*Carroll, 2003*; *Ghorbani et al., 2016*; *Zhang, 2016*). On the other hand, it is also necessary to apply the exponent "time" $T(e^{(T)})$ to obtain a linear result of the estimated survival time after the graft evaluation period.

In addition, this study allowed adjusting the Weibull risk function to estimate the probability of occurrence of negative events (graft mortality). To make this adjustment, the "SurvRegCensCov" library was used, which allows transforming the AFT model into the Weibull risk function, which is defined as:

$$h(t) = \lambda\gamma(te^{-\beta x})^{\lambda-1}e^{-\beta x} \tag{5}$$

Where: $h(t)$ is the risk function, $\lambda$ is the model form parameter ($\lambda > 0$), $\gamma$ is the scale parameter ($\gamma > 0$), $\beta$ is the explanatory variable coefficient, which can take values between positive and negative infinity (*Carroll, 2003*). Finally, $t$ is the average elapsed survival time in which graft mortality events occurred at some moment within the evaluation period (note that $t < T$), when the start of the first monthly evaluation is considered as the study origin. The values for the parameters $\lambda$, $\gamma$ and $\beta$ are estimated with the following equations, that imply the use of the parameters calculated for the above AFT model (Eq. 4):

$$\lambda = e^{-\alpha/\sigma} \tag{6}$$

$$\gamma = \frac{1}{\sigma} \tag{7}$$

$$\beta = \frac{-\delta}{\sigma} \tag{8}$$

In turn, the Weibull risk function allows obtaining the risk coefficient or ratio (hazard ratio, HR) of an explanatory variable ($x$), which measures a specimen's probability of death based on said variable, over a time ($t$) previously defined by the researcher. If $\lambda > 1$ the value of HR increases, while HR decreases if $\lambda < 1$ (*Carroll, 2003*; *Zhang, 2016*). If the value of the hazard ratio is 1.0, means that the risk of death is equal between treatments; if the value of the hazard ratio is less than one, it means a reduction in the risk of death in one of the treatments, while if the hazard ratio is greater than one, the risk of death increases in one of the treatments (*Carroll, 2003*; *Ghorbani et al., 2016*) The HR is determined by the following equation (*Igl, 2018*):

$$HR = \left(e^{-\beta}\right)^{\lambda-1} \tag{9}$$

The parameters $\beta$ and $\lambda$ of the HR equation (Eq. 9), are the same parameters calculated for the Weibull risk function (Eq. 5).

The Weibull risk function model uses dummy variables to predict the HR. To perform survival analysis on the R platform, the "censored" data is used to indicate living specimens, which are coded with a zero (0) (*Pohar & Stare, 2006*). The censored data are those specimens which are alive at the end of the evaluation, and are named that way because it is not possible to know precisely at what time they will experience the death event. Therefore, dead grafts at the end of the evaluation period were coded with a one (1). The donor-tree type (TDT) and scion position in the donor's tree crown (SPC) were coded as follows: Grafts made with scions from trees of the pure species (0) and grafts made with hybrid tree scions (1); grafts with scions from the upper third of the crown (0) and grafts with scions from the middle third of the crown (1).

## RESULTS

### Survival rate

Figure 3 shows the results of grafts survival during the first 4 months of evaluation. In the 1st month after grafting, the survival condition of the scion grafts from the middle third of

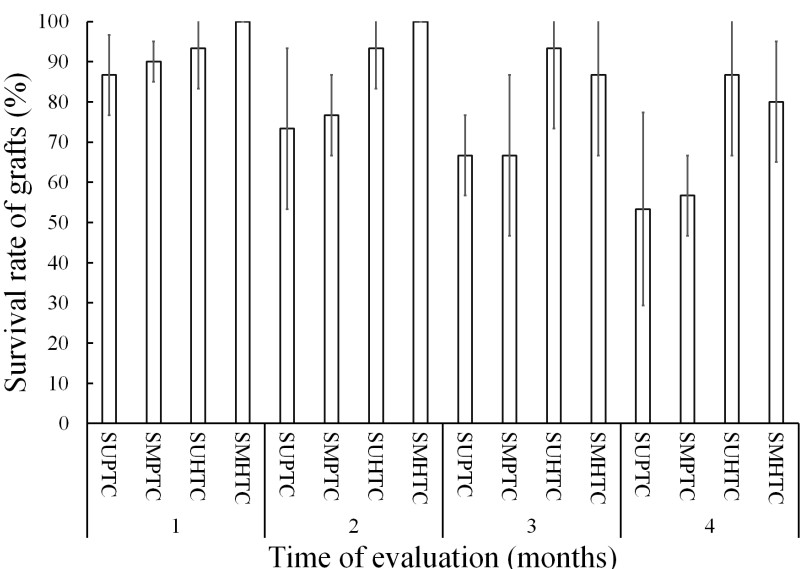

**Figure 3 Monthly survival rate in grafts with scions from the pure species *Pinus engelmannii* and from the *P. engelmannii* × *P. arizonica* hybrid for each treatment, in the first four months of evaluation.** Where: SUPTC, scion grafts from the upper third of the pure-species tree crown; SMPTC, scion grafts from the middle third of the pure-species tree crown; SUHTC, scion grafts from the upper third of the hybrid tree crown; SMHTC, scion grafts from the middle third of the hybrid tree crown. No significant differences were found between treatments in any separate month of evaluation. The whiskers represent the standard error of each treatment.

the crown of hybrid trees was total, while the treatment with lower survival was that of scion grafts from the upper third of the crown of trees of the pure species with 86.6%.

On the other hand, scion graft treatments from the middle third of the pure species tree crown and scion grafts from the upper third of the hybrid tree crown had 90.0% and 93.3% survival, respectively. In the first monthly evaluation no significant differences were found between treatments ($p = 0.390$).

In the 2nd month of evaluation, scion treatments from the upper third of the pure-species tree crown and scions from the middle third of the pure-species tree crown, showed a survival of less than 80%. In contrast, scion grafts from the middle third of the crown of hybrid trees survived in their entirety, while the survival in the treatment of scion grafts from the upper third of the crown of hybrid trees was 93.3%; even so, no significant differences were found ($p = 0.126$).

In the 3rd month of evaluation, survival decreased to 66.6% for both scion treatments of pure species-donor trees; the treatment with the highest survival was that of grafts made with scions from the upper third of the crown of hybrid trees with 93.3%. The survival of the scion grafts from the middle third of the crown of hybrid trees decreased from 100% to 86.6% in the third month of evaluation; however, the four treatments were statistically equal ($p = 0.128$).

Survival decreased in the four treatments in the fourth month of evaluation, when the best treatment was that of scion grafts from the upper third of the crown of hybrid trees,
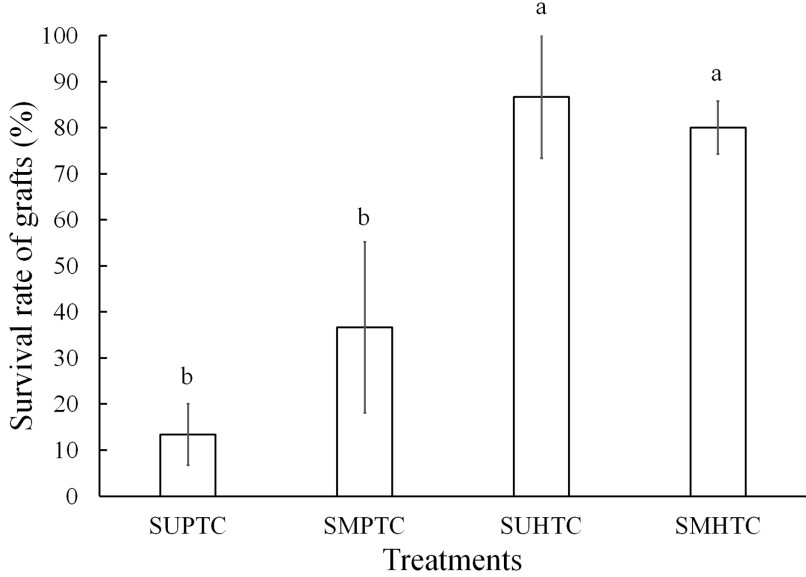

**Figure 4 Monthly survival rate in grafts with scions from the pure species *Pinus engelmannii* and from the *P. engelmannii* × *P. arizonica* hybrid for each treatment, in the fifth month of evaluation.** Where: SUPTC, scion grafts from the upper third of the pure-species tree crown; SMPTC, scion grafts from the middle third of the pure-species tree crown; SUHTC, scion grafts from the upper third of the hybrid tree crown; SMHTC, scion grafts from the middle third of the hybrid tree crown. Different letters indicate significant differences between treatments ($p < 0.0083$). The whiskers represent the standard error of each treatment.

with survival at 86.6%, however, the treatments continued without showing significant differences ($p = 0.132$).

Five months after grafting, survival varied from 86.6% to 13.3% among the treatments evaluated, with significant differences after the Bonferroni correction ($p < 0.0083$) between treatments. The grafts with better results occurred in the materials with hybrid tree scions (Fig. 4). In the two treatments where the effect of the position of the scion in the crown of the hybrid donor trees was separately evaluated, the results were statistically the same. On the other hand, the successful sprouting of the graft achieved with hybrid scions taken from both positions of the crown, was statistically superior to the grafting performed in the two treatments in which pure species donor tree scions were used. Significant differences were not found ($p \geq 0.0083$) when the comparison was made just between grafts made with scions taken from the two crown positions of pure species donor trees (Fig. 4).

## Pearson's correlation test and stepwise regression to adjust the logit and Weibull models

A Pearson's correlation test indicated that there is highly significant collinearity only between the height of the rootstock and height of the graft, so it was decided to exclude the RH variable in the adjustment of the Logit and Weibull models and include graft height, since the literature on *Pinus* grafts indicates that this last variable influences survival (*González Jiménez et al., 2017*). However, the stepwise regression applied to select variables

**Table 2 Stepwise regression to select variables to adjust the Logit model and the Weibull accelerated failure time model.**

| Variables | Number of variables | AIC |
|---|---|---|
| TDT, SPC, SL, SD, RD, GH, GD | 7 | 135.5 |
| TDT, SPC, SD, RD, GH, GD | 6 | 133.6 |
| TDT, SPC, SD, RD, GD | 5 | 132.4 |
| TDT, SPC, RD, GD | 4 | 131.1 |
| TDT, SPC, GD | 3 | 130.0 |
| TDT, GD | 2 | 129.2 |
| TDT | 1 | 129.0 |

Note:
 AIC, Akaike information criterion; TDT, Type of scion donor tree; SPC, Position of the scion in the donor tree crown; SL, Length of the scion; SD, scion diameter; RD, Diameter of the rootstock at the root crown; GH, Graft height; GD, Rootstock diameter at graft height.

**Table 3 Level of significance of the Logit model estimators with respect to death probability of grafts of trees of the pure species of *Pinus engelmannii* and the *P. engelmannii* × *P. arizonica* hybrids.**

| Parameter | Estimator | $\lvert z \rvert$ | $p < \lvert z \rvert$ |
|---|---|---|---|
| Intercept ($\alpha$) | −1.09 | 0.68 | <0.0001 |
| Coefficient of the "type of scion donor tree" ($\beta_1$) variable | 2.70 | 0.45 | <0.0001 |

to adjust the Logit and Weibull models, indicated that none of the graft variables was useful to describe the probability of survival significantly and not affected the probability of graft mortality (Table 2), since the only predictor variable that showed significant effect was the type of scion's donor tree.

## Odds ratio for graft survival according to the logit model

It was found in the Logit model that the intercept and the coefficient of the variable "type of scion donor tree" were highly significant ($p < 0.0001$) after the Bonferroni correction, in terms of the mortality of *P. engelmannii* grafts (Table 3). The probability $p(x_i)$ of graft survival rates for each type of scion donor tree, during the evaluation period, is illustrated in Fig. 5.

The probability dispersion diagram (Fig. 5) indicates a different decreasing rate of the probability of graft survival, between the analyzed hybrid trees and trees of the pure species trough time. In the probability diagram, it is possible to verify that due to the positive value of the $\beta_1$ coefficient of the "type of scion donor tree" predictor variable (Table 3), the probability of survival increases by increasing the predictor variable $x_i$ by one unit, that is to say when increasing from $x_0 = 0$ (trees of the pure species) to $x_1 = 1$ (hybrid trees). The probability of graft survival with pure-species tree scions ($p(x_0)$) was 25%, while the probability of graft survival with hybrid tree scions ($p(x_1)$) was 83.3% at the end of the 5 month evaluation period.

At the end of the evaluation, the number of live specimens were 50 grafts with scions from hybrid-donor trees and 15 grafts with scions from trees of the pure species;

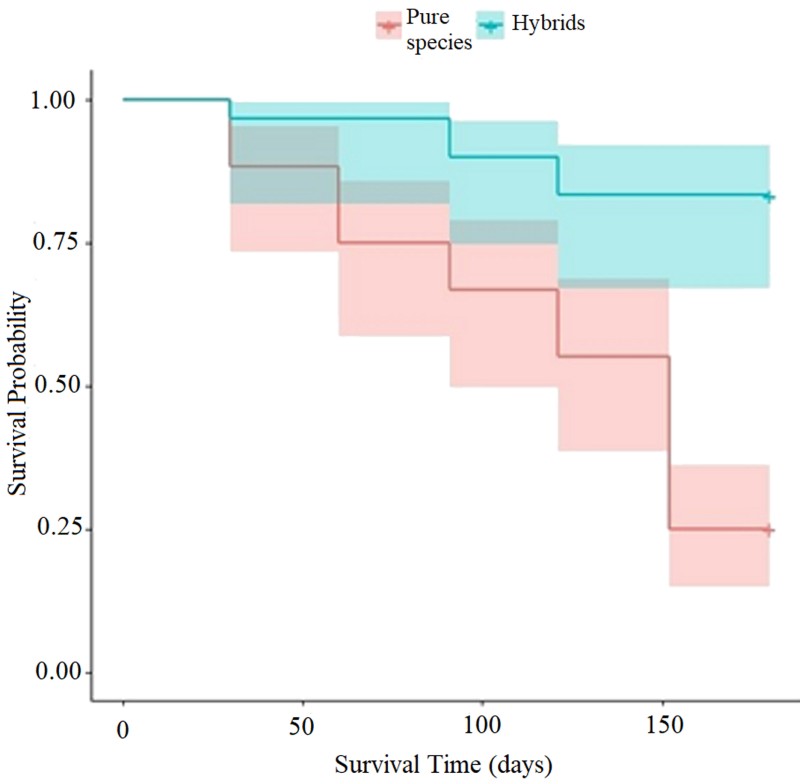

**Figure 5 Graft survival probability diagram (in logarithmic basis) with scions from trees of the pure species of *Pinus engelmannii* and from *P. engelmannii* × *P. arizonica* hybrids.**

meanwhile, there were 10 and 45 dead grafts with scions from hybrids and trees of the pure species, respectively. The estimated odds ratio (Eq. 3) for the variable "type of scion donor tree" was 15.0, which was obtained with the following operation, which uses the relative frequencies of live and dead grafts by treatment:

$$\text{Odds ratio} = \frac{a \times d}{b \times c} = \frac{(50/120) \times (45/120)}{(10/120) \times (15/120)} = 15.0$$

Note that because the numerator variable *a* represents the relative frequency of living grafts with scions from hybrid-donor trees, the result of the equation means that the odds ratio (proportion) of successful grafts made with scions from hybrid-donor trees, in the long run after the evaluation period, will be 15 to one with respect to grafts made with pure species-donor tree scions.

## Survival probability of *P. engelmannii* grafts according to the AFT model and Weibull risk function

The AFT model was highly significant for predicting the average survival time elapsed until the occurrence of the death event of one or more of the grafts that reached alive up until the end of the evaluation period. In addition, all parameters of the AFT model were highly significant even after the Bonferroni correction (Table 4).

**Table 4 Estimation of the Weibull accelerated failure time model parameters to predict the graft survival time.**

| Parameter | Estimator | \|z\| | $p < $ \|z\| |
|---|---|---|---|
| Intercept ($\alpha$) | 5.0456 | 61.92 | <0.0001 |
| Coefficient of the "type of scion donor tree" ($\delta$) variable | 1.0595 | 4.80 | <0.0001 |
| Scale ($\sigma$) | 0.5460 | −4.96 | <0.0001 |

**Table 5 Estimation of Weibull risk function parameters, to assess the risk of death in pine grafts, according to the type of donor tree.**

| Parameter | Estimator |
|---|---|
| Form parameter ($\lambda$) | $9.69 \times 10^{-05}$ |
| Scale parameter ($\gamma$) | 1.83 |
| Coefficient of the "type of scion donor tree" ($\beta$) variable | −1.94 |

According to the coefficient value $\delta$ of the "type of scion donor tree" variable, and when applying the exponent ($e^{(1.06)} = 2.88$), it is interpreted that when the $x$ variable takes the value of 1 (hybrid-donor trees), the survival time $T$ increases at a rate of 2.88 compared to what happens when $x = 0$ (pure species-donor trees). When developing the AFT model (Eq. 4), with the estimated parameters (Table 4) it was obtained that the estimated survival time for grafts with scions from pure species-donor trees is 156 days on average, while the graft survival time with scions from hybrid-donor trees is 450 days on average. In other words, it is estimated that the death of at least one graft would occur at 156 days in the case of scions from trees of the pure species and up to 450 days among the grafts of hybrid tree scions.

Depending on the $T$ results mentioned above, the difference in survival between treatments will increase as time $T$ also increases. The high statistical significance ($p < 0.0001$) of the two parameters ($\alpha$ and $\sigma$) and the coefficient $\delta$ shown in Table 4, allows accepting that the AFT model is significant and therefore, its parameters can be used to estimate the risk function (Eq. 5) and to estimate the hazard ratio (Eq. 9) of graft mortality, as a function of the type of scion donor tree. The parameters obtained for the Weibull risk function ($h(t)$) are shown in Table 5.

When applying Eq. (9), the hazard ratio at a time $t = 150$ days for the "type of scion donor tree" variable was 0.14 when referring to hybrid trees, while the hazard ratio in relation to scions donated by trees of the pure species is much higher (1.22). Whereby, the risk of graft death when using hybrid-donor tree scions is reduced by 86% ((1.0 minus hazard ratio) (100) = (1 − 0.14) (100) = 86). This risk of death, depending on the "type of scion donor tree" variable, is represented in the diagnostic diagram of the Weibull model (Fig. 6).

The cumulative risk of death of grafts with pure species-donor tree scions is greater during the entire evaluation period compared to graft survival when using hybrid tree scions. The cumulative risk of death of grafts with hybrid tree scions was only 0.14 at 5 months of evaluation. The differences are notorious in the risk of death between

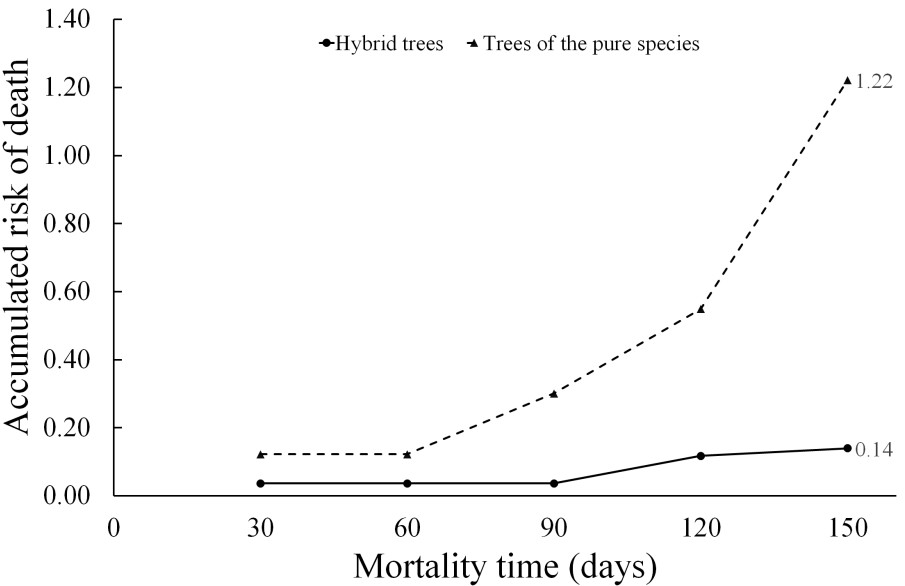

**Figure 6 Diagnosis diagram of the accumulated hazard ratio in the five month evaluation time, according to the Weibull regression.**

treatments, since the accumulated hazard ratio curves are not intercepted at any point in the diagram.

## DISCUSSION

In this side veneer grafting experiment a general survival of 54% (including all treatments) was obtained. Propagating *Pinus* and other coniferous species through grafting techniques has been viable in several experiments, obtaining survival results that range between 50% and 80% (*Lott et al., 2003*; *Hibbert-Frey et al., 2011*; *Almqvist, 2013a*). Interspecific grafting is an alternative to improve propagation results by this means, in species that have been difficult to graft by intraspecific methods (*Barnes, 2008*). In two of the four treatments evaluated in our research, survival of at least 80% was obtained; the two treatments with this survival were those of grafts made with scions from *P. engelmannii* × *P. arizonica* hybrids. In this study, the results are similar to those obtained by *Hibbert-Frey et al. (2011)*, who reported 86% survival in *Abies fraseri* (Pursh) Poir, in top cleft grafts at 4 months of evaluation. A similar success rate (53.8% at 6 months of evaluation) was obtained by *González Jiménez et al. (2017)* in side-veneer grafts of *Pinus leiophylla* Schiede ex Schltdl. *et* Cham. On the other hand, in a top cleft-grafting experiment of *Pinus sylvestris* L., a 75% survival was obtained in the first year after grafting (*Almqvist, 2013a*). Also, *Almqvist (2013b)* reported a survival of 84.7% 6 months after the grafting of *P. sylvestris*, using the top cleft-grafting technique. In side-veneer grafts of *P. engelmannii*, a 22.5% survival rate was reported at 6 months of grafting (*Pérez-Luna et al., 2019*).

In the present experiment, significant differences were observed between the graft of pure species-donor and hybrid-donor trees at 5 months of evaluation, with greater survival in the grafts of scions taken from hybrid trees. In the final evaluation (5 months) of our study, an 83% survival was observed in grafts made with *P. engelmannii* × *P. arizonica*

hybrid scions, while in grafts with scions taken from pure species *P. engelmannii*, a survival was achieved of only 25%. However, research on the use of *Pinus* hybrids for graft propagation is scarce; in top cleft grafts of *Pinus palustris* Mill. × *P. elliottii* Engelm. var. *elliottii* hybrids on *P. palustris* rootstocks, *Lott et al. (2003)* reported a 72% survival rate 1 year after grafting.

Although there were no significant differences in the first 4 months of evaluation, we observed that the survival percentage was always higher in the treatments with scions from hybrid-donor trees (Fig. 3). This suggests that the scions taken from hybrid trees may have a greater potential for graft adaptation and healing, than the scions taken from trees of the pure species, when both are grafted on rootstock from pure species *P. engelmannii*.

The importance of the study of forest areas that have experienced hybridization has be pointed out, especially in order to identify the advantages of using hybrids for adaptation to extreme environments in which trees of the pure species have less vigor, a solution that can be used in reforestation programs (*Zobel & Talbert, 1984*; *López Upton et al., 2001*; *Wachowiak et al., 2016*). Several studies also indicate that the vigor in plant species may be greater in hybrid offspring, at least in the first generation of hybridization (*Djidonou et al., 2016*; *Xu et al., 2016*; *Zhang et al., 2019*). *Ávila-Flores et al. (2016a)* in their work on hybrid stands of trees in Durango, México, in which the degree of hybridization between *P. engelmannii* × *P. arizonica* was studied, indicate that these hybrids may have the potential for adaptation in drought conditions.

Some authors consider that the presence of hybrids in seed stands decreases the genetic gain of the progeny (*Arnold & Hodges, 1995*; *Rieseberg & Carney, 1998*; *Wehenkel et al., 2017*). However, other authors recognize the importance of studying the effect of hybrids in breeding programs (*López Upton et al., 2001*; *Nunes et al., 2018*; C. Wehenkel et al., 2020, unpublished data). *Mabaso, Ham & Nel (2019)* reported that the *Pinus patula* Schltdl. & Cham. × *P. tecunumanii* F. Schwedtf. ex Eguiluz *et* J.P.Perry hybrids, showed little resistance to frost. However, *Nunes et al. (2018)* found better increments and adaptability of the hybrid *P. elliottii* var. *elliottii* × *P. caribaea* var. *hondurensis*, when planted in regions suitable for the establishment of the pure species *P. elliottii*, in Brazil. In this sense, and according to our results, although there are no differences between the growth and vigor of the scions' donor trees, progeny tests should be performed to determine the hybrids' potential for establishing ASO, for obtaining appropriate seeds for the production of plants in the nursery, which would serve for improving the establishment of commercial forest plantations in disturbed areas. On the other hand, it is advisable to carry out studies to determine the viability of the use of hybrids for vegetative propagation, provided there is evidence that hybrid trees show better adaptation and development characteristics than pure species-parent trees (*López Upton et al., 2001*; *Reig et al., 2019*).

In this work, only the effect of the scions from trees of the pure species of *P. engelmannii* and hybrids of *P. engelmannii* × *P. arizonica*, grafted onto pure species *P. engelmannii* rootstocks was evaluated, finding that the best combination was to graft hybrid scions onto pure species rootstocks. In a graft experiment by *Larix gmelinii* (Rupr.) Rupr. var. *japonica*, it was found that when using *L. gmelinii* var. *japonica* × *L. kaempferi* (Lamb.) Carr. hybrid rootstocks, survival was 92.6% in the first year of grafting, because the scions from trees of

the pure species were very compatible with hybrid rootstocks (*Kita et al., 2018*). In our study, a similar percentage of survival (83.3%) was found in grafts made with scions from *P. engelmannii* × *P. arizonica* hybrid trees on pure species *P. engelmannii* rootstocks (Fig. 4). *Marchal et al. (2017)* performed grafts on *Larix decidua* Mill. × *L. kaempferi* hybrid rootstocks using three types of scions: Hybrid scions from *L. decidua* × *L. kaempferi*, scions from the pure species *L. decidua* trees and scions from the pure species *L. Kaempferi* trees, and found that the grafts with pure-species tree scions from both species were the best. They concluded that the success in graft survival in the genus *Larix* was not favored by the use of scions from hybrid trees. This situation contrasts with the results found in the present study, where better survival was obtained in grafts made with hybrid tree scions.

In our study, we consider that a possible cause of the high mortality (and potential incompatibility) in the pure species grafts is a result of the high presence of resin channels in *P. engelmannii* trees. This conclusion was reached, based on the work of *Pérez-Luna et al. (2019)*, who found that a greater number of resin channels in the pure species *P. engelmannii* scions corresponded to a diminished survival of side-veneer grafts. In addition, *Yeaton, Yeaton & Waggoner (1983)* indicate that trees with fewer stomata in the needles are more resistant to stress; and *P. arizonica* contains fewer stomata in the needles than *P. engelmannii* (*García Árevalo & González Elizondo, 2003*), which could have affected the degree of compatibility in the graft between the *P. engelmannii* × *P. arizonica* hybrid scion and the pure species *P. engelmannii* rootstock.

In this work, no significant differences were found between the grafts with scions from the upper third of the crown compared to the middle third, neither in hybrid-donor trees nor in pure species-donor trees. In contrast, *Holst, Stanton & Yeatman (1963)*, found that the survival rate of *Pinus sylvestris* grafts was 96% when grafting scions from the top of the crown, while scion grafts from the bottom of the crown had a survival rate of 82%. Also, in *Abies fraseri* top cleft grafts, a 90% survival rate with scions taken from the top of the crown was achieved; results that were statistically different ($p < 0.05$) with respect to the survival rate achieved in grafts with scions from the bottom part of the crown, which ranged between 50% and 70% (*Hibbert-Frey et al., 2011*).

In our results, there were no significant differences between the different positions of the scions in the pure species tree crown ($p = 0.73$). However, a trend was observed for a greater survival in grafts with scions from the middle third of the crown, compared to the upper third (Fig. 4). Further studies are necessary to verify the possible effect of the scions' position on the tree crown on the success of grafting in *P. engelmannii* and other species of the same genus. *Rosier et al. (2005)* reported that the *Abies fraseri* scions from the lower zone of the crown were better for achieving the propagation of this species by rooting stem cuttings. Although grafts and cuttings are different vegetative propagation techniques, there are species like *Pinus leiophylla* that have been propagated by both methods, and some of the factors that have influenced their success are similar (*Cuevas Cruz, 2014*, *Cuevas-Cruz et al., 2015*).

The graft variables of the scions and rootstocks showed no significant effects on the probability of graft death in this study, as the stepwise regression revealed that none of the

graft variables significantly affected graft survival (Table 2). This may mean that the survival of grafts from pure species *P. engelmannii* and *P. engelmannii × P. arizonica* hybrids is not affected by the diameter and length of the scions and rootstocks, nor by the height and diameter of the section where the graft is tied. On the other hand, significant differences were found only in relation to the type of scion donor tree (Table 1).

However, *González Jiménez et al. (2017)* found that, in *Pinus leiophylla* side-veneer grafts, there was greater success when grafting at a height close to the base of the rootstock (5–10 cm) with a 53% survival rate. In another *P. leiophylla* grafting experiment, it was found that when the diameter of the rootstock was larger, the quality and development of the grafts were higher (*González Jiménez, 2017*). In future studies, it will be important to use rootstocks and scions of different ages and sizes to conduct studies on the effect of the dasometric characteristics of the rootstocks and scions on the grafting success of *P. engelmannii* and other species of *Pinus*.

Furthermore, satisfactory results were found in this study in the adjustment of the Logit model and the Weibull risk function to obtain the odds ratio and the hazard ratio respectively, which allowed estimating the graft survival and death probabilities depending on the "type of scion donor tree" variable. It should be noted that the use of survival models in research concerning commercial forest plantations or vegetative propagation of conifers has been very scarce (*Sigala Rodríguez et al., 2014*). The only recent work found in this regard is that of *Pérez-Luna et al. (2019)*, who reported that it was possible to estimate the hazard ratio for grafts with donor tree scions of different ages, with the adjustment of the Cox proportional hazards model.

Survival models were very useful for calculating the estimated average survival time of the grafts that remained alive until the end of the evaluation period. Furthermore, knowing the effect of the variables studied through the odds ratio and the hazard ratio will be very useful for decision making in subsequent research.

It is advisable to continue investigating the factors involved in the success or failure of graft propagation of *Pinus* species, at least those of greater economic or environmental importance, with a view to forming asexual and clonal seed orchards. The results obtained in this study can serve as a basis for future work in this area of research.

## CONCLUSIONS

The sprouting of grafts on pure species *P. engelmannii* rootstocks was better when using scions from *P. engelmannii × P. arizonica* hybrid-donor trees. By contrast, using scions from the trees of the pure species increased the probability of death of grafts of this species. The lower survival rate in grafts made with scions from trees of the pure species was noticeable from the third month of grafting, which may indicate that mortality in grafts with scions of the trees of the pure species occurs earlier than the mortality of grafts with hybrid tree scions.

The position of the scions in the crown of hybrid or pure species-donor trees did not show significant effects on the evaluated treatments. As a result, the success rate may be similar when grafting with scions collected from any part of the tree crown of the

analyzed species. Furthermore, the dimensions of the graft variables evaluated in the scions and grafted rootstocks seemed not have influenced the probability of graft survival.

The adjustment of the Weibull AFT model made it possible to predict that there is a greater probability of death when using scions of pure species *P. engelmannii* trees. Adjustment of the Logit model also allowed for obtaining the odds ratio for survival, in response to the aforementioned variable, observing that there is greater success in grafting when using scions taken from hybrid *P. engelmannii* × *P. arizonica* trees.

## ACKNOWLEDGEMENTS

The authors are grateful to M.C. Santiago Solís González for the support provided in harvesting the vegetative material used in the present study. We also thank Dra. Socorro González Elizondo for the support provided for the botanical identification of the studied species.

### Funding

This work was funded by the Consejo Nacional de Ciencia y Tecnología (CONACYT) (441054) and Consejo de Ciencia y Tecnología del Estado de Durango (COCYTED-12/02/18/265). The funders had no role in study design, data collection and analysis, decision to publish, or preparation of the manuscript.

### Grant Disclosures

The following grant information was disclosed by the authors:
Consejo Nacional de Ciencia y Tecnología (CONACYT): 441054.
Consejo de Ciencia y Tecnología del Estado de Durango (COCYTED-12/02/18/265).

### Competing Interests

Christian Wehenkel is an Academic Editor for PeerJ.

### Author Contributions

- Alberto Pérez-Luna conceived and designed the experiments, performed the experiments, analyzed the data, prepared figures and/or tables, authored or reviewed drafts of the paper, and approved the final draft.
- Christian Wehenkel conceived and designed the experiments, analyzed the data, authored or reviewed drafts of the paper, and approved the final draft.
- José Ángel Prieto-Ruíz conceived and designed the experiments, performed the experiments, authored or reviewed drafts of the paper, and approved the final draft.
- Javier López-Upton conceived and designed the experiments, authored or reviewed drafts of the paper, and approved the final draft.
- José Ciro Hernández-Díaz conceived and designed the experiments, analyzed the data, prepared figures and/or tables, authored or reviewed drafts of the paper, and approved the final draft.

## Data Availability

The raw data are available in the Supplemental Files.

## Supplemental Information

Supplemental information for this article can be found online at http://dx.doi.org/10.7717/peerj.8468#supplemental-information.

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
