# Peer review of "Survival of side grafts with scions from pure species Pinus engelmannii Carr. and the P. engelmannii × P. arizonica Engelm. var. arizonica hybrid"

_PeerJ, doi:10.7717/peerj.8468_

## Round 0.1 · original submission · Minor Revisions

Both reviewers agree on the relevance and quality of your work. However, they suggest several changes in the presentation of your work (e.g. inclusion of background information/ structure, etc.) and the revision of your references. As well, please review and use the correct terminology. E.g. is "dasometric variables" equivalent to "stand variables"? If this is the case, please use the more commonly used term.

Reviewer 1 ·

Basic reporting

The manuscript is relatively clear and relevant from the scientific point of view. The objective is relevant, but the purpose of morphological analysis should be explained earlier in the subsection, for example, in hypothesis, and incorporate in general objective. Some correction of language is needed.

Some tables and figures are described insufficiently making their interpretation incomplete.
Table 1. Define the value after the means. Is this SD? This is necessary to describe.
Fig. 2. The same comment: whiskers should be described.
Fig. 3. It is necessary to indicate the basis of the figure and describe what the scatter means in this case.

Experimental design

Some parts of the experimental design are not well explained. It is unclear, how many grafts were made, because the numbers in the manuscript does not correspond the experimental design. The authors wrote "120 grafts were made using Pinus engelmannii rootstock; 60 grafts were done with scions from hybrid-donor trees and the other 60 with scions from pure trees". But the second part of the experiment includes 20 scions from 2 parts of 6 trees (3 pure species and 3 hybrids). Isn't it 240 trees, 120 per species/hybrid variant? This should be explained.

The terms "high hybridization" and "degree of hybridization" should be clarified. Does this relate to F1, introgression or AFLP analysis? This is important, because scions were collected based on this information.

Validity of the findings

Some intermediate and final conclusions are doubtful.
It is impossible to draw a conclusion about the effect of the graft height on probability of survival from an experiment with a relatively uniform rootstocks and scions (L438-439). This would be possible if they had obviously different parameters. If they really had, this is unclear from the experiment description.

It is unclear, why the authors consider incompatibility as the probable cause of the graft death. As was fairly pointed out in the "Materials@Methods", there are many factors affecting the survival of grafts. This is especially noticeable against the fact that no biological differences between species and hybrids were described in the article, and we could not make any assumptions about the nature of the incompatibility.

L428-430 This is a trend here, and it’s normal to mention it, but the authors need to clearly indicate that this is a trend. The words "arithmetically higher" sound doubtful.
It is known, that in many species the crown part from which the scion originates is important not only for graft survival, but also for the graft growth form (Araucaria sp., Tectona grandis). In other species, this phenomenon is not observed, like in this manuscript. The biology of the species is very important in this case. The authors could name some of the distinguishing traits of P. engelmannii and P. arizonica that led to this result. This also could explain why P. engelmannii has low graft survival. Indeed, some species are best propagated by grafting, and some by cuttings. What about P. engelmannii? Maybe that is why it is so poorly propagated by grafting? And could it happen that P. arizonica is propagated better by grafting, so their hybrids have greater survival in grafts? All this doubts could be solved by the proper description of both species.

Additional comments

## Major Comments
The author have to mention both parent species when talking about hybrids, not only "hybrids of Pinus engelmannii".
The term "pure trees" is incorrect, it can be misinterpreted as a tree without grafting. Pure species is correct, this term is often used in high-ranked journals (Sujii et al., 2019; Baena-Díaz et al., 2018; Wu et al., 2018).
In the Discussion the main points of the results or first part of them should be briefly summarized before comparing them with others.
It is important to say how hybrids differ from pure species and whether there are advantages in their plantation planting. Do they grow better or not? The aspects of adaptation is well described, but morphology is also important. The main thing is to make it clear whether it makes sense to replace species with hybrids in specific cases of plantation planting.
It would be good to see in Materials@Methods main morphological characteristics of the species and hybrids, as well as the parameters of donor trees, including their approximate age and height.


## Specific Comments
L84-85 Please define the early stage of development and indicate the age of initiation of reproduction in the grafts.
L90 Cross is correct term, not cross-linking.
Fig. 1 Please do not use the same letters, if there are no differences anywhere, just write it in the legend. The whiskers need to be identified.
L297 Since several samples are being compared, the Bonferroni correction should have been applied in all cases of comparison, and this is specified in the methodology. Is there a reason, why this is mentioned in this case?
L303 P-value varies between the text and the figure.
Do not provide a reference to the same figure or table in the same subsection several times, one reference is enough.

Reviewer 2 ·

Basic reporting

1) The article is well written, using clear and technical correct text.
2) The introduction presents a good background referred to the situation of forest species in Mexico and the problem of the loss of forest area as posing a study that can help implement the solution to this problem. However, the main objective is the study of the response to the graft and the various factors that might influence and determine the success of this propagation, and on this issue there are some suggestions:
As specified in lines 79-82, there are several factors involved in the success of coniferous grafting, including graft compatibility, as the author explains in one of the references (Pera Luna et al., 2019). The introduction should be improved at this point in more detail. Between lines 83 and 87, there are some references to this problem but refer to other species.
If the exit of the graft depends of several factors, including compatibility, this aspect must be resolved at the beginning of the approach, to ensure the results obtained in the work.
I suggest that some references related to the graft behavior of the species involved in this study be included, to focus the work and the results obtained in a successful graft response, without the interference of a possible incompatibility response.
In line 92, some references say the opposite of the sentence, the sentence should be rewritten to know better where the lack of studies on the use of hybrids in conifers is.
In line 94-96 there are references to studies in fruits and vegetables, but the three references are two of tomato and one of melon, not fruits.

3) Table 2 should be explained in more detailed. First they explain that exclude the height of the rootstock, and then conclude that there are no dasometric variables affecting mortality of grafts, referred to Table 2, but do not explain how it reached this conclusion

Experimental design

1) The grafting process has practical applications in various crop production and analysis of the mechanisms involved it is relevant to many interesting questions in the development of plants.
2) The research question is well defined. The main objective of this work is to elucidate the implication of some factors involved in the success of the graft. However, the authors apply a methodology based on statistical models to assess the probability of occurrence of successful grafts. But there is no mention in the state of the art of previous studies using this methodology, or the novelty that means applying these models. I suggest the inclusion of some reference concerning the application of these models in other studies, to better understand how this study helps fill the research question.
3) The investigation has been conducted rigorously and to a high technical standard.
4) The methods are described with enough information for another researcher to reproduce.
However, in some cases there are too many details; e.j the location by coordinates of the institute where the grafts were made is not relevant.
Some sentences are not well formulated: line 111: "Therefore, at the time of grafting, the rootstock was 2.5 years old." This date is not a consequence of the treatment explained above.
Until line 121, it seems that the selection of plant material between hybrids and pure lines was made only by visual selection. But on line 122, it is revealed that the selection was made after the use of molecular markers of type AFLP. I suggest rewriting the sentence.

Some photography would be useful to understand the dasometric variables, for example referring to the length of the stem. Root Height? And graft height, with respect to what? It is not clear what these variables refer to.
The assessment of the models is explained very widely. I am not familiar with this methodology, so I cannot assure you that it has been applied correctly.

Validity of the findings

1) The novelty of the results is clear, then on the assumption made, results are discarded that do not provide answers and study new possibilities arise with the applied models. In my opinion, they should focus on the novelty of the results in this regard on the findings of work.
2) The conclusions are well stated linked to the original research. Only the reference to the ‘incompatibility’ should be considered. It is not the same the survival rate of the grafts and the graft incompatibility response. The graft incompatibility response is related to the genetic of the both combinations, and the survival of the graft combinations, and the studies related to this hypothesis, are not involved with the graft response.
As no compatibility studies have been done, my suggestion is that the conclusions should not refer to this response, and they should be mention in the introduction, previous studies that have been done in this type of combinations
The conclusions do not refer to the models used to estimate odds ratios and hazard ratio to know the effects of the variables studied, and they are an important part of the results. It should be mentioned in the conclusions.

The article is novel and addresses a topic of interest for the propagation of species. It is well structured and activities are well directed. However, revisions on some topics are required to strengthen the hypothesis and the studies performed. It is necessary to differentiate compatibility problems from graft success, as well as provide prior information on compatibility work in these species. And to place the results of models in a context to previous work and the novelty that it implies.

Additional comments

no comment

---

## Round 0.2 · accepted · Accept

Thank you very much for addressing all reviewers' comments in your corrected version of the manuscript.